# WaAgents: A Waterfall-Inspired Framework for Effective Multi-Agent Collaboration

## Abstract

Large Language Models (LLMs) have revolutionized the construction of multi-agent systems for complex problem solving, leveraging their prowess in natural language understanding for semantic parsing and intent recognition, alongside robust logical reasoning for intricate task execution. Despite these advances, prevailing LLM-based multi-agent frameworks suffer from a critical shortfall: the absence of explicit, predefined stage segmentation. This leads to pervasive information redundancy in inter-agent communications, manifesting as irrelevant discussions without focused topics, and exacerbates decision conflicts in free-discussion paradigms, where agents of equal status deadlock over divergent opinions, ultimately hindering effective resolutions. To address these limitations, we introduce WaAgents, a novel multi-agent collaboration framework inspired by the Waterfall Model in Software Engineering. WaAgents delineates the problem-solving process into four sequential, interdependent stages: Requirement Analysis, Design, Implementation, and Reflection. In the Requirement Analysis stage, Requirement Analysis Agents parse user intents to produce a structured task specification, facilitating downstream processing. Designer Agents in the Design stage then employ this specification to decompose the task into granular sub-tasks, systematically assigning them to dedicated Worker Agents. During Implementation, each Worker Agent executes its sub-task through targeted operations and computations. Anomalies trigger the Reflection stage, where Error Analysis Agents diagnose root causes, distinguishing design from implementation errors, and enact precise repairs, ensuring iterative refinement without disrupting workflow integrity. This stage-driven, highly structured workflow provides each agent role with explicit, concentrated objectives, which substantially mitigate information redundancy. Furthermore, by strictly enforcing the predefined flow, WaAgents fundamentally eliminates the decision conflicts inherent to free-discussion, thereby ensuring the coherence and effectiveness of the entire solution process. Empirical validation across challenging benchmarks, including mathematical reasoning and open-ended problem solving, confirms the efficacy and marked superiority of the WaAgents framework.

## 1 Introduction

Large Language Models (LLMs) have demonstrated exceptional capabilities in both natural language understanding and logical reasoning, establishing a foundational core for constructing multi-agent systems geared toward complex problem-solving (Li et al., 2024). Specifically, their robust understanding enables LLM-based agents to perform semantic parsing and intent recognition of user requirements, while their reasoning faculty empowers them to execute sophisticated tasks.

Recent research has explored various multi-agent collaboration frameworks based on LLMs (He et al., 2025). A prevalent approach involves context-based communication, where LLM-based agents, often through role-playing, collaboratively derive solutions via multi-turn dialogue. For instance, methods such as Camel (Li et al., 2023) adopt a two-agent dialogue strategy, where an initial user request guides the interaction between the two agents to complete the task. Another category of frameworks, typified by AgentVerse (Chen et al., 2024b), utilizes a strategy of negotiation and communication among multiple agents, primarily simulating human group brainstorming processes to resolve problems (Wang et al., 2023). Through continuous interaction, agents contribute to a final

decision, returning the result upon consensus. However, these existing methods exhibit significant limitations when tackling complex problem-solving (A detailed example showing the limitations of existing methods can be seen in section A.1.):

- Lack of Explicit Stage Demarcation: The absence of clear, staged partitioning during the problem-solving process hinders effective collaboration. Without well-defined discussion topics for each step, agents are prone to generating a large volume of information redundancy (Han et al., 2024) (i.e., information irrelevant to the problem's solution), severely impeding efficiency.

- Vulnerability to Decision Conflicts: The reliance on a free-speech discussion model for problem-solving is particularly problematic when multiple agents have divergent opinions. Since agents operate on an equal footing, they may become entrenched in their own viewpoints, leading to decision conflicts (Sun et al., 2025) that make it difficult to arrive at a correct and effective resolution efficiently. Furthermore, ambiguity or hallucination during the exchange can cause agents to enter endless arguments or discussion loops (Huang et al., 2025) without the ability to self-diagnose the root cause of the issue, ultimately preventing a solution from being found (Xi et al., 2025).

To address these challenges, this paper proposes a Waterfall-Inspired Framework for Effective Multi-Agent Collaboration—referred to as WaAgents. The core conceptual underpinnings of WaAgents are derived from the engineering methodology of Software Engineering (SE), which emphasizes the systematic division of complex system development into distinct, sequentially executed stages (Flora & Chande, 2014). Specifically, WaAgents adopts the classic Waterfall Model (Royce, 1987) from SE to segment the multi-agent problem-solving process into four sequential stages: Requirement Analysis, Design, Implementation, and Reflection.

- Requirement Analysis: The Requirement Analysis Agent(s) focus exclusively on understanding and parsing the user's intent. The output of this stage is a structured task specification that is easily interpretable by subsequent agents.

- Design: The Designer Agent receives the structured task specification and systematically decomposes the complex task into a series of sub-tasks, which are then assigned to the respective Worker Agent(s).

- Implementation: Each Worker Agent executes specific operations and computations based on the sub-task it has been assigned.

- Reflection: Should an anomaly occur during the Implementation Stage, the system transitions to the Reflection Stage. The Error Analysis Agent is tasked with diagnosing the error based on the anomaly information, localizing the error (identifying if the fault lies in the Design Stage or the Implementation Stage), and then initiating the corresponding repair and correction procedures.

This stage-driven collaboration model provides each stage with explicit objectives, enabling the agents to concentrate their efforts on the task at hand, which substantially reduces information redundancy (Rasal & Hauer, 2024). Moreover, by strictly adhering to the predefined workflow, the framework fundamentally eliminates decision conflicts among agents that may arise from a free-discussion mode, thereby ensuring the coherence and effectiveness of the solution process. [1]. The main contributions of the paper are:

- In terms of theoretical methodology, we propose the WaAgents multi-agent collaboration framework based on the SE waterfall model, which effectively addresses the limitations of existing LLM-driven multi-agent systems in complex problem-solving, such as information redundancy and vulnerability to decision conflicts, through stage division.

- In terms of technical details, in the Reflection Stage, the Error Analysis Agent enables precise error attribution (locating faults in the Design or Implementation Stage) during implementation anomalies and initiates targeted repair processes, substantially improving the robustness of multi-agent systems in handling complex problems.

---

[1]This paper involves all the source codes and specific experimental data, which can be found in the anonymous repository through https://anonymous.4open.science/r/WaAgents-8887/README.md

- In terms of experimental evaluation, results across multiple challenging benchmarks, including mathematical reasoning and open-ended problem solving, validate the efficacy and superiority of the WaAgents framework. The framework demonstrates significant performance improvements over existing multi-agent methodologies, particularly in task completion rate and the consistency of the output with the user's initial requirements.

## 2 RELATED WORKS

**LLM-based Autonomous Agents** Since the concept of artificial intelligence was first introduced, researchers have been committed to developing agents capable of solving problems autonomously without human intervention (Xi et al., 2025). In recent years, the emergence of LLMs has opened up new pathways for enhancing the capabilities of such agents (Barua, 2024). Notably, agent architectures like ReAct (Yao et al., 2023b) and Reflexion (Shinn et al., 2023), which integrate reasoning with action, have laid important groundwork for the field. Building on this foundation, researchers have been advancing agent capabilities from various angles (Xi et al., 2025). For example, the work by Yao (Yao et al., 2023a) focuses on improving agents' task planning and decomposition abilities through Chain-of-Thought, while Schick (Schick et al., 2023) explores how agents can invoke external tool APIs to meet specific task requirements. These efforts collectively aim to enhance the adaptability and practicality of agents in complex real-world scenarios, in order to cope with increasingly diverse user needs.

**Multi-agent System** In the real world, teams of people working together are often more efficient and better at handling complex problems than individuals working alone (Park et al., 2023). This observation has inspired many researchers to explore multi-agent systems based on LLMs. The main goal of such systems is to assign different roles and responsibilities to multiple LLM-powered agents, allowing them to work together to solve complex tasks (Hong et al., 2024). Currently, most research focuses on building multi-agent collaboration frameworks that rely on dialogue mechanisms (Guo et al., 2024). For example, CAMEL (Li et al., 2023) is a conversational framework that uses role-playing to enable collaboration between agents, but it only supports interaction between two agents. To extend this type of dialogue-based collaboration to more agents and improve problem-solving capabilities, later studies have developed more advanced group discussion mechanisms (Becker, 2024). For instance, AutoGen (Wu et al., 2024) allows multiple interactive agents to solve problems through group discussion, while AutoAgents (Chen et al., 2024a) introduces an observer role to better guide the discussion process and prevent confusion. However, the performance of these systems heavily depends on the quality of the discussions among agents. If disagreements arise and no consensus can be reached, the conversation may fall into loops, which can reduce overall collaboration efficiency.

## 3 THE DETAILS OF WAAGENTS

This framework strictly divides the problem-solving workflow into four sequential stages: Requirement Analysis, Design, Implementation, and Reflection, as shown in Fig 1. Each stage produces a verifiable intermediate result. This staged approach ensures an orderly and traceable collaboration process, thereby significantly enhancing the overall effectiveness of the system.

### 3.1 REQUIREMENT ANALYSIS

User requirements often suffer from issues such as semantic ambiguity and unclear intent, which can lead to outcomes generated by multi-agent collaboration deviating from the user's true intentions. Existing research is increasingly recognizing the importance of clarifying user requirements. To address this, we first introduce a *Clarification Question Expert* to identify vague or ambiguous points in the original requirements and generate a set of clarification questions for the user to answer. After the user responds to these questions one by one, we leverage a *Requirement Optimization Expert* to refine the requirements based on the clarification feedback, producing content that is more logically precise while eliminating ambiguous and insufficient descriptions.

Gherkin is a structured, natural-language-based requirements specification language. It uses keywords like Feature, Scenario, Given, When, and Then to describe requirement behaviors. Compared

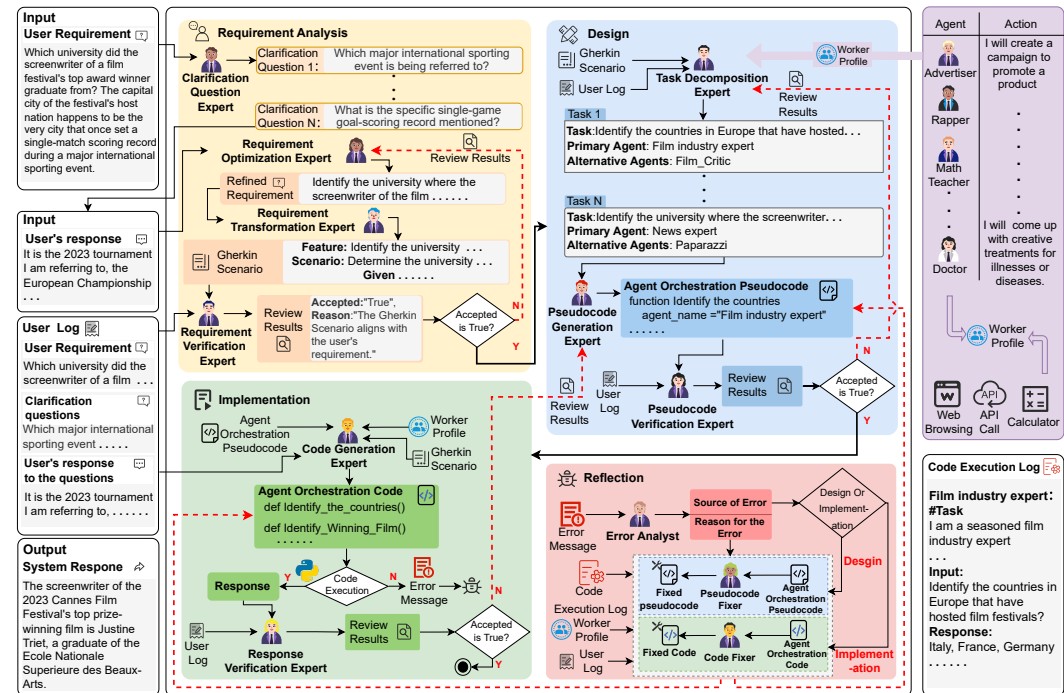

Figure 1: The overview of our WaAgents approach

to informal natural language, this structured format offers higher readability and comprehensibility for agents. The AgileGen framework proposed by (Zhang et al., 2025). also indicates that converting user requirements into structured descriptions using Gherkin effectively enhances semantic consistency between requirements and generated code. The clarified and optimized user requirements are then converted into a standardized Gherkin format. This structured representation not only removes the inherent ambiguity of natural language but also provides clear task steps for subsequent stages, thereby supporting more accurate task decomposition and code generation.

To fully ensure alignment with user intent, we further validate the Gherkin-formatted requirements using a User Log. This log, composed of the initial user requirements, the clarification questions, and the user's responses, comprehensively reflects the user's actual needs. If the validation passes, it indicates a high consistency between the Gherkin requirements and user intent, and the process proceeds to the Design stage. Otherwise, the *Requirement Verification Expert* provides reasons for the failure, and the Requirement Optimization Expert subsequently re-optimizes the requirements based on this feedback. A more detailed prompt can be found in the appendix A.2.1.

## 3.2 DESIGN

Although the Requirement Analysis stage effectively addresses the challenge of ambiguous user requirements, directly generating code from Gherkin requirements still presents issues. At its core, this approach remains reliant on a single agent tackling complex problems directly. When code generation errors occur, this method makes identifying and rectifying faults considerably more difficult. To address this issue, we adopt the principle of separation of concerns by dividing the code generation process into two distinct stages: Design and Implementation. This division enables our framework to focus separately on task logic design and concrete code realisation, thereby significantly reducing the complexity of individual stages, enhancing the precision of error diagnosis, and improving the maintainability of the entire workflow.

Within the Design stage, we introduce the *Task Decomposition Expert*. This agent takes the clarified Gherkin requirements as input and performs task decomposition and worker selection based on a *Worker Profile*. Decomposing complex tasks into smaller, more manageable subtasks is a core strat-

egy for handling requirement complexity. It not only reduces the cognitive load on individual agents but also facilitates error localization and repair. Related research has confirmed its effectiveness in multi-agent systems (Park et al., 2023; Sumers et al., 2023). The Worker Profile is a knowledge base composed of various specialized agents and tools (Deng et al., 2023) within the *Worker layer*. In this study, these worker agents are sourced from a curated collection selected from a high-star open-source project on GitHub. This collection covers multiple domains such as education, finance, and news. We manually fine-tuned them, enabling capabilities like web search and API calls, to ensure they adapt to our framework. The Task Decomposition Expert strictly follows the Single Responsibility Principle, breaking down complex user requirements into a series of subtasks. Each subtask focuses on a single, clear objective and is assigned to the most suitable worker agent from the Worker profile for execution.

Next, the *Pseudocode Generation Expert* is responsible for converting the set of subtasks into pseudocode. Pseudocode is a natural-language-based intermediate representation for describing algorithmic logic. Due to its high alignment with the logical flow of tasks, it effectively expresses task sequences in a form that is easy for agents to understand and process. Furthermore, as an abstraction layer between design intent and concrete code implementation, pseudocode plays a key role in effectively decoupling design from implementation. It allows the framework to independently design and optimize task logic without getting prematurely bogged down in the syntactic details of programming languages. To ensure the generated design aligns with the user's original intent, the *Pseudocode Verification Expert* validates the generated pseudocode. This expert compares the pseudocode against the User Log from the requirements analysis stage to verify that its semantics and logic fully conform to the user's true requirements. If the verification passes, the process proceeds to the implementation stage. If it fails, the verification Expert provides Review Results to the Pseudocode Generation Expert, which then regenerates or revises the pseudocode accordingly. More detailed prompts can be found in the appendix A.2.2.

## 3.3 IMPLEMENTATION

In the Implementation stage, we introduce the *Code Generation*, which is responsible for converting the pseudocode from the Design stage into executable Python code. Each subtask is implemented as a specific function, and worker agents are mapped as units that execute these functions. Using code as the method for multi-agent collaboration offers clear advantages over approaches based on informal dialogue: code can precisely and unambiguously represent task flows and provide verifiable intermediate results for complex tasks. Existing research, such as the " Code-As-Language " paradigm proposed by Schick et al. (2023), also indicates that translating natural language intents into code behaviors can significantly improve the accuracy and verifiability of task execution.

The generated Python code is executed directly using an embedded Python interpreter. If the code runs successfully, the output is validated by the *Response Verification Expert* to ensure it semantically aligns with the user's requirements. Should validation fail, it indicates that the outcome fails to meet user requirements at the semantic level. The issue may stem from Design flaws within the pseudocode, such as incorrectly defined input-output specifications for the agent. Consequently, the Pseudocode Generation Expert must revise the pseudocode based on the Review Results. If an error occurs during execution, the system captures complete error information and transfers it, together with the corresponding Python source code and Code Execution Log, to the Reflection stage. This provides full context for subsequent error diagnosis and repair. More detailed prompts can be found in the appendix A.2.3.

## 3.4 REFLECTION

If the code execution fails, the process proceeds to the Reflection stage. First, the *Error Analyst* analyzes the error information to diagnose the root cause and determine whether the error originated in the Design stage or the Implementation stage. This step is crucial as it prevents indiscriminate fixes and ensures that corrective actions are targeted. For example, a Design-level error might manifest as pseudocode that omits output specifications for a worker agent, leading to abnormal responses, while an Implementation-level error could be an incorrect reference to an undefined variable during code generation.

If the error is determined to originate from the Design stage, the *Pseudocode Fixer* initiates a repair process. It corrects the flawed pseudocode logic by consulting the complete Code Execution Log, the Worker Profile, and the Reason for the Error. If the error is confirmed to stem from the Implementation stage, it indicates that the assigned agents and subtask set are correct, but there is an issue in the code generation. In this case, the *Code Fixer* performs the repair based on the Code Execution Log, the Worker Profile, and the diagnosed error reason. More detailed prompts can be found in the appendix A.2.4.

# 4 EXPERIMENTS AND EVALUATION

## 4.1 EXPERIMENTAL SETUPS

**Benchmarks** To comprehensively evaluate the performance of WaAgents, we selected three benchmark datasets covering different task types, namely MGSM (Shi et al., 2022), BFCL (Patil et al.), and MT-Bench (Bai et al., 2024). Specifically, we adopt the English subset of MGSM proposed by Shi et al. (Shi et al., 2022), which originates from GSM-8K (Cobbe et al., 2021) and contains multiple elementary school-level mathematical problems. This benchmark is mainly used to assess the logical reasoning and numerical calculation capabilities of the model. BFCL (Patil et al.) focuses on the tool-use capabilities of LLMs in real-world scenarios, covering single-/multi-turn dialogue and serial/parallel function calling. It can effectively evaluate the task planning and collaborative consistency of multi-agent frameworks. MT-Bench (Bai et al., 2024) consists of 80 high-quality multiturn questions, covering 8 core scenarios including writing, role-playing, information extraction, reasoning, mathematics, coding, STEM knowledge, and humanities / social science knowledge. Each scenario contains 10 manually designed challenging questions, which are used to evaluate the coherence of multi-turn dialogue and the instruction-following ability of the LLM.

**Evaluation Metrics** Depending on the task characteristics of each dataset, we employ a combination of automatic[2] evaluation and subjective human evaluation for experimental evaluation. For the MGSM dataset, since its tasks involve closed mathematical problem solving, we use *Success Rate* as the evaluation metric. The Success Rate is defined as the proportion of samples where the model's output completely matches the standard answer, which directly reflects the model's mathematical reasoning and calculation accuracy. For the open-ended tasks in BFCL and MT-Bench datasets, we use the *Win Rate* as the core evaluation metric to quantify the performance advantage of WaAgents over baseline methods such as Camel (Li et al., 2023), Agent-Verse (Chen et al., 2024b), and AutoAgents (Chen et al., 2024a). The calculation formula is defined as $(Num_{win} + 0.5 * Num_{equal})/Num_{total}$, in which, $Num_{win}$ refers to the number of samples where WaAgents outperforms the baseline methods, $Num_{equal}$ refers to the number of samples where WaAgents performs equally to the baseline methods, and $Num_{total}$ refers to the total number of samples. The determination of the Win Rate is implemented through two evaluators: FairEval (Sah et al., 2025) and HumanEval. FairEval reduces evaluation biases through multiple strategies to align well with human judgments. In the BFCL scenario, the win/loss is determined based on the function calling success rate and user consistency. In the MT-Bench scenario, it is determined based on usefulness, reliability, accuracy, and detail level. As for HumanEval, we recruit 3 master students with research experience in multi-agent systems to conduct manual scoring according to the same dimensions as FairEval, and the win/loss is determined based on the scoring results. These annotators received unified training to clarify scoring criteria (e.g., output quality, consistency with requirements). During the evaluation, we hide the method names to avoid bias, and the final score is the average of the 3 evaluators' ratings to reduce subjective deviation. Through the two evaluators, we not only verify the reliability of win rate determination via automatic evaluation (FairEval) but also compensate for the limitations of automatic tools in the depth of semantic understanding.

**Foundation Large Language Models** To verify the adaptability and performance stability of our WaAgents, we select two base LLMs with different capability levels: GPT-3.5-Turbo and GPT-4o. Among them, GPT-3.5-Turbo represents an LLM with a medium capability level, while GPT-4o represents an LLM with a relatively high capability level. We implement the calling of these two LLMs by accessing OpenAI's API. Meanwhile, to ensure the consistency and reproducibility of

---

[2]The prompt for the scoring agent are provided in the appendix. A.3

experimental results, we uniformly set the model's Temperature parameter to 0, so as to avoid result fluctuations caused by excessively high randomness.

## 4.2 Comparing with the Baseline Methods

Table 1: Success rates comparison on the MGSM dataset

| Model | WaAgents | Single | Camel | AgentVerse | AutoAgents |
|-------|----------|--------|-------|------------|------------|
| GPT-3.5-Turbo | 64.8% | 33.6% | **84.4**% | 83.6% | 12.8% |
| GPT-4o | **97.6**% | 61.6% | 95.2% | 94.8% | 89.2% |

Table 2: Win Rate of WaAgents against other methods on the BFCL and MT-Bench

| Dataset | Evaluator | Model | Single | Camel | AgentVerse | AutoAgents |
|---------|-----------|-------|--------|-------|------------|------------|
| BFCL | FairEval | GPT-3.5-Turbo | 54.0% | 58.0% | 41.0% | 84.5% |
| | | GPT-4o | 63.5% | 95.5% | 74.5% | 57.5% |
| | HumanEval | GPT-3.5-Turbo | 56.5% | 52.5% | 44.5% | 80.0% |
| | | GPT-4o | 64.5% | 92.5% | 72.0% | 56.5% |
| MT-Bench | FairEval | GPT-3.5-Turbo | 46.3% | 45.0% | 51.2% | 88.7% |
| | | GPT-4o | 62.6% | 53.1% | 52.5% | 60.3% |
| | HumanEval | GPT-3.5-Turbo | 48.7% | 46.2% | 50.8% | 86.1% |
| | | GPT-4o | 58.4% | 52.7% | 51.1% | 58.5% |

In this subsection, we compare WaAgents with four baseline methods, namely Single Agent, Camel, AgentVerse, and AutoAgents. For the Single Agent method, we directly invoke LLMs to generate results without any collaboration or process optimization, relying solely on the inherent capabilities of the LLM to complete tasks. The experimental results can be seen in Table 1 and Table 2.

In the MGSM dataset, WaAgents shows efficient adaptability to both LLMs, with particularly prominent advantages on high-performance models. As shown in Table 1, when using the GPT-3.5-Turbo model, WaAgents achieves a Success Rate of 64.8%, which is significantly higher than the 33.6% of Single Agent and 12.8% of AutoAgents, and only lower than the 84.4% of Camel and 83.6% of AgentVerse. This is because the task reasoning chains of MGSM are relatively short, typically requiring only 2–3 steps of reasoning logic. Camel's fixed "user-assistant" dual-agent dialogue can quickly decompose simple problems, and the multi-agent system of AgentVerse can quickly cover basic problem-solving paths. Although WaAgent's structured process has shown optimization value compared with Single Agent, the efficiency of medium-capability models in handling procedural tasks (such as pseudocode generation and error detection) is limited, and the potential of the process has not been fully unleashed. However, when using the GPT-4o model, WaAgent's Success Rate increases to 97.6%, significantly outperforming all baseline methods. This indicates that high-performance LLMs can fully exploit the value of WaAgents' workflow, while the limitations of baseline methods become more apparent. Camel lacks dynamic task decomposition ability, making it prone to reasoning gaps when faced with multi-step tasks. The negotiation redundancy of AgentVerse occupies the context window, leading to the omission of key calculation steps. While the AutoAgents' feedback mechanism can only fix surface-level errors and cannot avoid deviations in the initial problem-solving ideas, resulting in inferior performance compared to WaAgents.

As shown in Table 2, on the BFCL dataset, we observe that WaAgents achieves its highest Win Rate against AutoAgents, with FairEval at 84.5% and HumanEval at 80.0% when using the GPT-3.5-Turbo model. This is because AutoAgents relies on conversational feedback for behavior optimization, which tends to introduce ambiguity in instruction invocation, whereas WaAgents' "pseudocode-to-executable code" workflow precisely aligns with the syntactic and logical requirements of function calls. WaAgents exhibited the lowest win rate when competing against AgentVerse, with 41.0% on FairEval and 44.5% on HumanEval. The reason lies in the fact that AgentVerse's negotiation mechanism occasionally forms local advantages during step adjustment. However, such advantages only manifest in simple scenarios involving models with moderate capabil-

ities. When compared with Camel and Single Agent, WaAgents achieved comparable or slightly better performance, with FairEval scores of 58.0% and 54.0%, respectively. On GPT-4o, WaAgents' win rate against Camel increases to 95.5% (FairEval). High-performance models significantly enhance WaAgents' call planning capability, enabling it to automatically generate pseudocode snippets. In contrast, Camel's natural language conversations still suffer from ambiguity, preventing it from accurately matching tool-calling requirements. The win rate against AgentVerse increases to 74.5% (FairEval), as the redundancy in AgentVerse's negotiation process becomes more pronounced on high-performance models. The win rate against AutoAgents decreases to 57.5% (FairEval), since AutoAgents' error intervention ability improves under GPT-4o, allowing it to quickly detect issues such as missing parameters. However, due to the inherent limitations of its conversational feedback mechanism, it still cannot match the precision of WaAgents' structured workflow.

On the MT-Bench dataset, WaAgents' workflow advantages contribute to its increasing competitiveness when using high-performance models. With GPT-3.5-Turbo, WaAgents achieves a high Win Rate against AutoAgents, with a FairEval score of 88.7%. However, its win rates against Camel and AgentVerse are relatively close, at 45.0% and 51.2% respectively on FairEval. This is because MT-Bench tasks are more based on human-like interactive naturalness. Camel's dual-agent role-playing dialogue aligns more closely with human conversational conventions, while AgentVerse's multi-agent negotiation enriches dialogue details through role supplementation. In contrast, WaAgents' ability to optimize coherence is limited on medium-capacity models. WaAgents' Win Rate against Single Agent is moderate (FairEval: 46.3%), as the native dialogue capability of Single Agent already meets basic interactive needs, and WaAgents' procedural optimization does not yet yield a significant advantage in this context. When evaluated on GPT-4o, WaAgents' Win Rate against Single Agent increases to 62.6% (FairEval), as the high-performance model improves the WaAgents' error correction capability during the reflection stage. WaAgents also shows slight improvements in Win Rates against Camel and AgentVerse, with FairEval scores of 53.1% and 52.5%, respectively. In contrast, WaAgents' Win Rate against AutoAgents declines to 60.3% (FairEval).

## 4.3 ABLATION EXPERIMENTS

To systematically evaluate the necessity and contribution of the core stages in WaAgents, we design three variant methods by removing the Requirement Analysis stage (w/o RA), the Design stage (w/o Design), and the Reflection stage (w/o Reflection), respectively. These three variants are applied to three datasets, and we calculated their Success Rates on MGSM as well as the Win Rates of the complete WaAgents against each variant on the BFCL and MT-Bench datasets. In this experiment, we use GPT-4o as the foundation LLM. The experimental results are presented in Table 3.

Across all tasks, the complete WaAgents achieves significantly higher win rates over the w/o RA variant, with its advantages showing cross-task consistency. On the MGSM dataset, the complete WaAgents achieves an accuracy of 97.6%, while the w/o RA variant only achieves 70.4%. On BFCL, the Win Rates of the complete WaAgents over w/o RA are 56.5% on FairEval and 58.0% on HumanEval. Similarly, on MT-Bench, the Win Rates are 54.4% on FairEval and 57.2% on HumanEval. These results indicate that the Requirements Analysis stage serves as the directional guarantee for WaAgents. It can extract key task constraints, such as implicit calculation conditions in MGSM, function parameter requirements in BFCL, and dialogue objectives in MT-Bench. Without this stage, the framework tends to deviate in execution direction due to misunderstandings of task requirements, thereby leading to a decline in performance.

The complete WaAgents also outperforms the w/o Design variant across all datasets, and the stage's value is more prominent in structured tasks, highlighting its role in optimizing task execution logic. On the MGSM dataset, the complete WaAgents' Success Rate is 9.4% higher than the w/o Design variant's 89.2%. This is because the pseudocode generated by the Design stage prevents logical confusion in multi-step reasoning. On the BFCL dataset, the win rates of the complete WaAgents over the w/o Design variant reach 70.5% (FairEval) and 73.5% (HumanEval). In MT-Bench, the win rates reach 65.6% in FairEval and 62.7% in HumanEval, demonstrating that the stage's structural planning for dialogue enhances information coherence, while its absence tends to result in fragmented responses.

On the MGSM dataset, the w/o Reflection variant's 70.0% success rate is 39.4% lower than the complete WaAgents, indicating that the Reflection stage is essential for correcting calculation errors and

logical flaws. On the BFCL dataset, the complete WaAgents achieves win rates of 72.5% (FairEval) and 71.0% (HumanEval) over the w/o Reflection variant, demonstrating the stage's role in verifying whether function call results meet user requirements. On the MT-Bench dataset, the Win Rates achieve 68.1% on FairEval and 70.3% on HumanEval. These results demonstrate the important role of the Reflection stage in verifying the validity of invocation results and handling exceptions.

Table 3: Ablation study of WaAgents stages across three datasets

| Dataset | Evaluator | WaAgents | w/o RA | w/o Design | w/o Reflection |
|---------|-----------|----------|--------|------------|----------------|
| MGSM | Success Rate | 97.6% | 70.4% | 89.2% | 70.0% |
| BFCL | FairEval | - | 56.5% | 70.5% | 72.5% |
|  | HumanEval | - | 58.0% | 73.5% | 71.0% |
| MT-Bench | FairEval | - | 54.4% | 65.6% | 68.1% |
|  | HumanEval | - | 57.2% | 62.7% | 70.3% |

## 5 LIMITATIONS AND DISCUSSION

**Limitations of WaAgents** Although WaAgents effectively improve the efficiency of multi-agent collaboration in solving complex problems, their design still has certain limitations. The current framework relies on the ability of agents to generate pseudocode during the design stage and execute code during the implementation stage. This dependency means that its performance is closely tied to the capabilities of the underlying large language model. Experimental results show that when using relatively weaker models (such as GPT-3.5-turbo), WaAgents may perform even worse than baseline methods. This is mainly because weaker models tend to produce pseudocode and code with insufficient logical rigor and accuracy, making it difficult to clearly and precisely express the problem-solving steps, which ultimately affects the framework's execution. Additionally, the experiments in this study were only validated using two models, GPT-3.5-turbo and GPT-4o, which somewhat limits the generalizability of our conclusions. The adaptability of the WaAgents framework to large language models of different sizes and architectures still needs further investigation. In the future, we plan to apply the framework to a wider variety of large language models to comprehensively evaluate its robustness and broad applicability.

**Coverage of the Datasets** The evaluation in this study is primarily based on three datasets: MGSM (focusing on mathematical reasoning), BFCL (focusing on tool usage), and MT-Bench (covering various scenarios such as writing, role-playing, and coding). Although these datasets cover several important areas and can effectively assess the multi-agent framework's capabilities on specific tasks, their scenarios and problem types are still insufficient to fully simulate the complex, ambiguous, and dynamic needs presented by users in the real world. Real-world problems are often more open-ended and involve the integration of knowledge across different fields. Therefore, the current experimental results have limitations in proving the general applicability of WaAgents. An important direction for future work is to extend the evaluation to broader and more complex domains, such as software development and programming, to further validate its generality.

## 6 CONCLUSIONS

This study introduces WaAgents, an innovative framework for multi-agent collaboration. Inspired by the classic waterfall model in software engineering, WaAgents systematically divides the problem-solving process into four distinct stages: requirement analysis, design, implementation, and reflection. Comprehensive experimental results demonstrate that WaAgents outperforms both single-agent models and other multi-agent frameworks across a variety of complex tasks involving reasoning, comprehension, and tool usage. Ablation studies further validate the critical contribution of each stage within the framework. By integrating software engineering principles, WaAgents pioneers new pathways for multi-agent collaboration. We are confident that this work paves the way for tackling broader task scenarios and advancing the development of assistive AI.

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

# A APPENDIX

## A.1 LIMITATIONS OF EXISTING METHODS

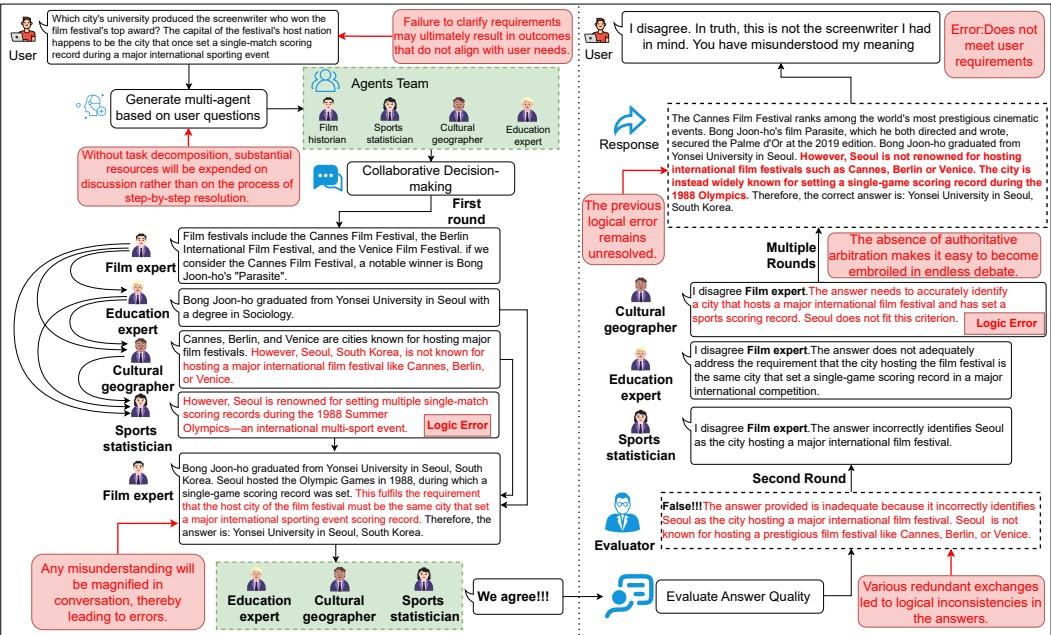

Figure 2: Flowchart of Existing Methods for Problem Resolution

Existing multi-agent collaboration methods based on dialogue mechanisms have been widely studied. However, these methods heavily rely on the quality of discussion among agents, often leading to circular debates and difficulty in reaching effective decisions. As shown in Fig 2, we use a specific example to illustrate the common limitations of such methods. The user's question is: "Which city's university produced the screenwriter who won the film festival's top award? The capital of the festival's host nation happens to be the city that once set a single-match scoring record during a major international sporting event.". This question contains several potential ambiguities. For example, the "international sports event" could refer to different types, such as football or track and field. If the task is solved directly without clarifying these needs, the result may easily deviate from the user's true intention.

In the example, the multi-agent team did not break down the task effectively but started a collaborative discussion directly. The film expert and education expert correctly pointed out that Bong Joon-ho, the director and screenwriter of Parasite, won the Palme d'Or at the Cannes Film Festival, and that he graduated from Yonsei University in Seoul. However, the cultural geographer mistakenly assumed that the " city where the screenwriter's university is located " must be the same as the " city where the film festival was held " leading to a chain of logical errors. The sports statistician then argued based on this information, stating that Seoul set a single-game scoring record during the 1988 Summer Olympics, which matched the description in the question. Thus, the team concluded that the answer was " Yonsei University in Seoul, South Korea. "

However, the team failed to recognize the core logical relationship in the question: the city that hosted the film festival should itself be the one that set the record in the sports event. The goal was to start from that city and identify the screenwriter and the university they graduated from. Although the reviewer noticed this logical error during the evaluation, it was not corrected after multiple rounds of discussion. As a result, the final response did not meet the user's needs. This case shows that without structured task decomposition and a clear collaboration process, multi-agent dialogue systems struggle to handle complex reasoning tasks that involve multiple layers of logic.

## A.2 PROMPT OF EACH STAGE

### A.2.1 REQUIREMENT ANALYSIS

---

**Clarification Question Expert**

---

You are a clarification question expert.
Your task is to review a natural language user requirement and determine whether it contains any ambiguities or missing information that should be clarified before implementation.

user requirement:
{{requirement}}

In multi-agent collaboration processes, ambiguous requirements frequently lead to ambiguities in intent. Therefore, to minimise such ambiguities, clarification questions must be posed in response to vague and non-explicit queries.
Below are the meanings of aspect for the query.
1. event: The action that the query requires.
2. status: The status of the object that the query requires. The modifier include adjectives, verbs, quantifiers, and adverbs.
3. type: The type of the object that the query requires. The noun or proper noun is the modifier. The modifier is Python built-in data type, such as "byte", "float", "char", "boolean", "double", etc
4. purpose: Purpose contains purpose clauses, which are employed to highlight the driving forces behind specific actions. The words "to," "in order to," and "so that" are used to start canonical purpose clauses.
5. condition: The conditio n of the query.

For each unclear or ambiguous aspect above, write several specific clarification question, you should keep your questions concise and focused.
output as the following format:
```

1. {question1}
...
```

If the requirement is fully clear and unambiguous.
output as the following format:
```

None
```

Please note that, you should only output the clarification questions or "None" if the requirement is fully clear and unambiguous in ``` ```., don't output any other words.

---

**Input:** {{User Requirement}}        **Output:** {{Clarifying Questions}}

---

Figure 3: The prompt of Clarification Question Expert

**Clarification Question Expert** During the Requirement analysis stage, we first introduce a Clarification Question Expert. Its core function is to analyze the user's original requirements, identify any ambiguous or unclear parts, and generate targeted clarification questions. The prompt design for this agent is shown in Figure 3. In our design, we primarily refer to the requirement clarification dimensions proposed by the KPL method, which include: Event, State, Type, Purpose, and Condition. The Event dimension clarifies the specific action the user wants to perform. The state defines the condition of the relevant objects when the action occurs. Type categorizes the action or the objects involved. The purpose aims to uncover the user's underlying goal or reason for the action. Condition specifies the prerequisites that must be met for the requirement to be valid. Furthermore, if the initial

requirements are already clear and unambiguous, the Clarification Question Expert will not generate any clarification questions.

---

**Requirement Optimization Expert**

You are a requirement optimization expert.
You are given:
A natural language user requirement
A list of clarification questions related to that requirement (based on aspects like event, status, type, purpose, condition)

user requirement:
{{requirement}}

clarification questions:
{{clarification_questions}}

Your task:
Carefully review the clarification questions.
For each question, infer a reasonable answer based on context or general logic.
Based on these inferred answers, rewrite and expand the original requirement to be:
Clear and unambiguous
Logically complete
Suitable for implementation

Output Instruction:
Only output the final refined requirement, do not include any answers or intermediate reasoning.

**Input:** {{User Requirement}}
     {{Clarifying Questions}}     **Output:** {{Refined Requirement}}

---

Figure 4: The prompt of Requirement Optimization Expert

**Requirement Optimization Expert** After the user provides responses to the clarification questions, the Requirement Optimization Expert integrates the original user requirements with the clarification feedback to refine the initial requirement description. This process aims to eliminate ambiguities in the original requirements while supplementing key details, thereby generating a more complete and precisely articulated set of user requirements that are easier for downstream agents to accurately understand. The prompt design for this agent is shown in Figure 4

**Requirement Transformation Expert** Gherkin is a structured requirements specification language built on natural language grammar. It uses keywords (such as Feature, Scenario, Given, When, and Then) to organize and describe feature behaviors, ensuring high readability and machine interpretability. User requirements are converted into the standardized Gherkin format. This structured representation not only eliminates the ambiguity often found in free-form natural language but also provides clear task sequences and data dependencies. As a result, it supports accurate subtask planning and reliable code generation in later stages. The related prompt is shown in Figure 5.

**Requirement Verification Expert** As shown in the Figure 6, the input includes the processed Gherkin scenarios and the user's original requirements. Specifically, the Requirement Verification Expert compares the expected behavior extracted from the Gherkin scenarios with the original user requirements. If the verification is successful, the process moves to the design stage. If it fails, the review result is sent back to the Requirement Transformation Expert to regenerate the requirements.

---

### Requirement Transformation Expert

You are a requirement transformation expert.
Given the following user requirement:
{{refined_requirement}}

Please generate Gherkin based on the natural language. Use the standard structure of "Feature", "Scenario", "Given", "When", and "Then". Be concise,
accurate, and reflect the user's intent faithfully. Only return the Gherkin code block, no explanations.

---

**Input:** {{Refined Requirement}}          **Output:** {{Gherkin Scenario}}

---

Figure 5: Requirement Transformation Expert

### A.2.2 DESIGN

**Task Decomposition Expert** Task Decomposition Expert focuses on breaking down the structured Gherkin requirements into smaller subtasks based on the single-responsibility principle. Each subtask concentrates on one specific functional goal. These subtasks are then assigned to the most suitable worker agent from the Worker profiles. The related prompt is shown in Figure 7.

**Pseudocode Generation Expert** The goal of this agent is to convert the set of subtasks into clear and logical pseudocode. Because pseudocode closely follows the logical flow of the tasks, it makes it easier for subsequent agents to understand the execution order and function dependencies. As shown in Figure 8, the input for this agent includes the Gherkin scenario, the task plan, the refined requirements, and the Agent Documentation(Worker Profile). When generating the pseudocode, the Pseudocode Generation Expert only uses the primary worker agent assigned to each subtask in the subtask plan to build the pseudocode.

### A.2.3 IMPLEMENTATION

**Code Generation Expert** The agent converts the pseudocode generated in the design stage into executable Python code. This Python code is then directly executed by an embedded Python interpreter. This agent uses the Gherkin scenarios, refined requirements, pseudocode, and Agent Documentation as input to generate the executable code, As shown in Figure 10.

**Response Verification Expert** It primarily checks whether the response generated after code execution matches the user's requirements. As shown in the figure, the input includes the Gherkin requirements, the refined requirements, and the response produced by the code executed through multi-agent collaboration,As shown in Figure 11.

### A.2.4 REFLECTION

**Error Analyst** The Error Analyst performs error analysis based on the error messages generated during code execution. This process has two main goals. First, it aims to determine whether the error originated from the design stage or the implementation stage. Design-stage errors are typically caused by unreasonable subtask planning that leads to conflicts. For example, failing to standardize the output format can result in structural issues like missing data fields or mismatched return formats. In contrast, implementation-stage errors are related to the specific construction of the code, such as type mismatches or incorrect parameter concatenation. Second, the Error Analyst generates a concise yet informative error description. This description serves as a detailed record and explanation of the identified problem, clearly stating its root cause. It provides the necessary context for the subsequent Repair Expert, aiding in more accurate and targeted modifications to the code or design,,As shown in Figure 12.

**Requirement Verification Expert**

You are a requirement verification expert.
You are given:
A user requirement in natural language
The same requirement expressed in Gherkin format
A system response from an Agent or application

Your task:
Use the original user requirements to determine whether the Gherkin-formatted requirements fully satisfy the original user requirements.

Input:
Requirement:
{{requirement}}

Gherkin Format:
{{requirement_gherkin}}

Output Format:
Return a JSON object in the following format:
```json
{
  "accepted": true or false,
  "reason": "If not accepted, provide a brief explanation here. If accepted, this can be null."
}
```
If the response is valid and fully satisfies the requirement, return:
```json
{
  "accepted": true,
  "reason": "null"
}
```

**Input:** {{Refined Requirement}}
    {{Gherkin Scenario}}

**Output:** {{Verification Result}}

Figure 6: Requirement Verification Expert

**Task Decomposition Expert**

You are a task decomposition expert. Given a set of Available Agents with descriptions and a user requirement expressed in Gherkin format, your task is to:

Break down the Gherkin-described requirement into a list of clear, actionable sub-tasks.

For each sub-task, identify one "Primary Agent" and multiple "Alternative Agents" that can fulfill the same or similar functionality.

Only select Agents from the provided list of Available Agents, and the selected Agent names must be exactly consistent with those in the list. Do not invent or assume Agents.

Return your answer strictly as a JSON-formatted string in the following format:

```
[
  {
    "task": "description of subtask",
    "Primary Agent": "name_of_primary_agent_from_list",
    "Alternative Agents": ["alternative_agent_1", "alternative_agent_2"]
  }
]
```

Available Agents:

{{Agent Documentation}}

User Requirement (in Gherkin):

{{requirement_gherkin}}

User Requirement (in plain text):

{{requirement}}

Instructions:

Use only the Agent names from the list provided above. Keep task descriptions short but meaningful. Every Agent entry must be consistent with its described capabilities. Output only valid JSON. No explanation or comments.

**Input:** {{Agent Documentation}}  {{Gherkin Scenario}}  {{Refined Requirement}}   **Output:** {{Task Plan}}

Figure 7: Task Decomposition Expert

918
919
920
921
922
923
924
925
926
927
928
929
930
931
932
933
934
935
936
937
938
939
940
941
942
943
944
945
946
947
948
949
950
951
952
953
954
955
956
957
958
959
960
961
962
963
964
965
966
967
968
969
970
971

**Pseudocode Generation Expert**

You are a pseudocode generation expert.
Based on the following three inputs:
A list of available Agent and their documentation
The user's requirement written in natural language
The same requirement written in Gherkin format
Your task is to generate clear pseudocode using only the provided proxies, ensuring that the subsequent executable code generated will fulfill user requirements through logical orchestration of tasks.
requirement:
{{requirement}}

Gherkin Format:
{{requirement_gherkin}}

Agent_documentation:
{{Agent_documentation}}

Task_plan:
{{Task_plan}}

Guidelines:
The pseudocode must be modular: define multiple high-level abstract functions，each representing an Agent interaction or logical step.

Include a main function (or equivalent) that represents the overall orchestration logic.

The pseudocode should be:
Language-neutral
Implementation-agnostic
Highly abstract, avoiding any real names, syntax, or data formats
All logic must follow the structure implied by the Gherkin scenario.
Use only the Agents from the provided documentation.
Output only the pseudocode block, no explanation or extra text.
Avoid situations where executable code returns directly without passing through the Agent.
Note：that all operations must be implemented based on the agent rather than Python-wrapped functions,for example:re,json,.

Output Structure Example (Abstract Pseudocode Format):
```

function Step_A with requirement:
    Select the agent_name and call the large model with task_requirement.
    return result_X

function Step_B with an input based on the preceding text or requirement:
    Select another the agent_name and call the large model.
    return result_Y

function Main:
    output_X = Step_A
    output_Y = Step_B
    perform final processing or return result

**Input:** {{Refined Requirement}}    {{Gherkin Scenario}}
        {{Agent Documentation}}   {{Task Plan}}

**Output:** {{Pseudocode}}

Figure 8: Pseudocode Generation Expert

```
Pseudocode Verification Expert

You are a pseudocode verification expert.
You are given:
A user requirement in natural language
The same requirement expressed in Gherkin format
A system response from an Agent or application
Your task:Use the original Requirement alongside Gherkin Format requirement to determine
whether the pseudocode fully satisfies all requirements.
Input:
Original Requirement: {{requirement}}  Gherkin Format: {{requirement_gherkin}}
Pseudocode: {{pseudocode}}
Output Format:
Return a JSON object in the following format:
{
  "accepted": true or false,
  "reason": "If not accepted, provide a brief explanation here. If accepted, this can be null."
}
If the response is valid and fully satisfies the requirement, return:
{
  "accepted": true,
  "reason": "null"
}
```

**Input:** {{Refined Requirement}}   {{Gherkin Scenario}}
{{Pseudocode}}   **Output:** {{Verification Result}}

Figure 9: Pseudocode Verification Expert

**Pseudocode Fixer** If the error originates from the design stage, the Pseudocode Fixer will take over. As shown in the prompt in Fig 13, to support the repair process, we provide it with comprehensive contextual information. This includes the Worker profile assigned to each subtask during the design stage, the original pseudocode, and the error description generated by the Error Analyst.

**Code Fixer** If the error originates from the implementation stage, it indicates that the task plan from the design stage is sound, and the error lies primarily in the construction of the executable code. As shown in Fig 14, we provide the Gherkin requirements, agent documentation, the error description, and the problematic code as contextual prompts to guide the Code Fixer in fixing the issue. The repair process may involve correcting syntax errors, adding type conversions, or fixing improper parameter usage.

A.3   PROMPT OF FAIREVAL

1026
1027
1028
1029
1030
1031
1032
1033
1034
1035
1036
1037
1038
1039
1040
1041
1042
1043
1044
1045
1046
1047
1048
1049
1050
1051
1052
1053
1054
1055
1056
1057
1058
1059
1060
1061
1062
1063
1064
1065
1066
1067
1068
1069
1070
1071
1072
1073
1074
1075
1076
1077
1078
1079

**Code Generation Expert**

You are a code generation expert.
You are given:
A natural language description of the user's requirement
A Gherkin description of the user's requirement
A set of Agent_name documentation
A high-level pseudocode representation of the solution logic
A call method for making LLM calls

Your task is to convert the given pseudocode into fully functional Python code that fulfills the user's requirement using the specified Agent_names, ensure the final result returned by the code meets the user's requirements.

Natural language user requirement:
{{requirement}}
Gherkin format user requirement:
{{requirement_gherkin}}
Agent Documentation:
{{Agent_documentation}}
Pseudocode:
{{pseudocode}}

Use Python 3 with standard libraries, and onvert each function in the pseudocode into a corresponding Python function with real request logic.

**Input:** {{Gherkin Scenario}}    {{Refined Requirement}}    **Output:** {{Executable code}}
{{Agent Documentation}}  {{Pseudocode}}

Figure 10: Code Generation Expert

**Response Verification Expert**

You are a response verification expert.You are given:
A user requirement in natural language
The same requirement expressed in Gherkin format
A system response from an Agent or application

Your task:Use both natural language and Gherkin format to determine whether the response is relevant to the requirement.You don't need to judge whether the response is correct; it just needs to meet the requirements.

Input:
Requirement: {{requirement}}
Gherkin Format: {{requirement_gherkin}}
Response: {{response}}

Output Format:
Return a JSON object in the following format:
{
  "accepted": true or false,
  "reason": "If not accepted, provide a brief explanation here. If accepted, this can be null."
}
If the response is valid and fully satisfies the requirement, return:
{
  "accepted": true,
  "reason": "null"
}

**Input:** {{Gherkin Scenario}}    {{Refined Requirement}}    **Output:** {{Verification Result}}
        {{Response}}

Figure 11: Response Verification Expert

**Error Analyst**

You are an expert software engineer and debugging analyst
You are given:
A natural language user requirement
A Gherkin-style behavior specification
A pseudocode implementation, meant to fulfill the requirement
A generated Python code from that pseudocode
A runtime error message
One or more Agent documentation entries

Your task:
Analyze the error and determine where the root cause lies. The error can be in one of the following categories:

Design Layer (Pseudocode-Level Mistake)
This means the pseudocode itself is flawed. Possible causes include:
Incorrect Agent Selection: The Agent used in the pseudocode does not meet the actual need.
Incorrect Agent Usage Pattern: The Agent is generally correct, but used in the wrong way.
Logical Flaw: The sequence or logic of operations doesn't match the requirement or Gherkin steps.

Implementation Layer (Code Generation Mistake)
This means the Python code is incorrect  despite the pseudocode being valid. Possible causes include:
Incorrect syntax
Incorrect translation of steps
Misused variables, formatting issues, or Agent misapplication that was not present in the pseudocode

User Requirement: {{refined_requirement}}   Gherkin Specification: {{requirement_gherkin}}
Pseudocode: {{pseudocode}}   Python Code: {{executable_code}}
Runtime Error Message:  {{error_message}} Agent Documentation: {{Agent_docs}}

Output Format:
```json
{
  "error_aspect": "Design",  // or "Implementation"
  "error_description": "Explain the reason for the error. Include your analysis and reference to
pseudocode or code. Be specific: was it a wrong Agent? A logic flaw? A code-level syntax
issue? Make your reasoning clear and justified."
}
```

**Input:** {{Refined Requirement}}   {{Gherkin Scenario}}
{{Pseudocode}}   {{Executable Code}}
{{Error Message}}   {{Agent Documentation}}

**Output:** {{Error Aspect}}
{{Error Description}}

Figure 12: Error Analyst

## Pseudocode Fixer

You are a senior system designer and software architect.

You are given the following information:
A natural language user requirement
A Gherkin specification (Given-When-Then format) that defines the expected behavior
The pseudocode that was originally written to satisfy the requirement
The generated Python code from that pseudocode
A runtime error message indicating a failure during code execution
A set of  Agent documentation entries

Your task:
Revise the pseudocode to fix the design-level mistake that caused the code to fail at runtime.

In particular:
The pseudocode may contain an incorrectly selected Agent, Incorrect task flow, or a flawed logic flow
Your job is to identify and fix the root cause in the pseudocode

You may choose to:
Replace the Agent with a better one from the docs
Change how the Agent is called (e.g., different parameters, options, input structure and input user querys)
Fix the logical steps to match what the runtime requires

You do not need to produce runnable code.
You only need to return a corrected version of the pseudocode that is:
Aligned with the original requirement and Gherkin specification
Free from design errors that would cause runtime failure

User Requirement: {refined_requirement}   Gherkin Specification: {requirement_gherkin}

Current Pseudocode: {pseudocode}   Generated Python Code: {executable_code}

Error Description: {error_description}   Agent Documentation: {Agent_docs}

Make sure the new pseudocode:
Corrects the mistake that caused the runtime failure
Remains logically coherent and complete
Uses appropriate Agent and proper usage patterns as defined in the documentation
Matches the intent of the original user requirement and Gherkin steps

**Input:** {{Refined Requirement}}   {{Gherkin Scenario}}
{{Pseudocode}}   {{Executable Code}}         **Output:** {{Repaired Pseudocode}}
{{Error Report}}   {{Agent Documentation}}

Figure 13: Pseudocode Fixer

1242
1243
1244
1245
1246
1247
1248
1249
1250
1251
1252
1253
1254
1255
1256
1257
1258
1259
1260
1261
1262
1263
1264
1265
1266
1267
1268
1269
1270
1271
1272
1273
1274
1275
1276
1277
1278
1279
1280
1281
1282
1283
1284
1285
1286
1287
1288
1289
1290
1291
1292
1293
1294
1295

### Code Fixer

You are an experienced Python developer and software debugger.

You are given:
A natural language user requirement
A structured Gherkin specification that defines the expected behavior
A Python implementation that was generated from pseudocode
A runtime error message indicating failure during execution
An error description from a prior error analysis step, identifying the problem as an Implementation Layer issue
One or more Agent documentation entries relevant to the logic

Your task:
Fix the Python code so that:
It executes without runtime errors
It aligns with the intent of the user requirement and Gherkin behavior
It uses the appropriate Agents correctly (parameters, usage patterns, etc.)
It preserves the core logic that was intended by the original pseudocode, modifying only what is necessary to correct the implementation mistake

User Requirement: {{Refined Requirement}}    Gherkin Scenario: {{Gherkin Scenario}}
Python Code: {{executable_code}}   Runtime Error Message: {{error_message}}
Error Description: {{error_description}}   Agent Documentation: {{Agent_docs}}

Return only the corrected Python code.

Ensure:
The fix is minimal but sufficient to make the code run correctly
The code reflects the correct logic expected by the user and the Gherkin specification
All Agent calls are valid and correctly used (check documentation)
```python
{repaired_code}
```

**Input:** {{Refined Requirement}}   {{Gherkin Scenario}}
{{Executable Code}}  {{Error Message}}                   **Output:** {{Repaired Code}}
{{Error Description}}   {{Agent Documentation}}

Figure 14: Code Fixer

---

**FairEval**

You are a helpful and precise assistant for checking the quality of AI assistants' answer performance.

You are given:
A user's question
The answer provided by Assistant 1
The answer provided by Assistant 2

Input:
Question: {{question}}
The Assistant 1's response: {{answer_1}}
The Assistant 2's response: {{answer_2}}

We would like to request your feedback on the performance of two AI assistants in response to the user question displayed above.
Please rate the helpfulness, relevance, accuracy, level of details of their responses.
Each assistant receives an overall score on a scale of 1 to 10, where a higher score indicates better overall performance.

Please first provide a comprehensive explanation of your evaluation, avoiding any potential bias and ensuring that the order in which the responses were presented does not affect your judgment. Then, output two lines indicating the scores for Assistant 1 and 2, respectively.

Output with the following format:
Evaluation evidence: <evaluation explanation here>
The score of Assistant 1: <score>
The score of Assistant 2: <score>

---

**Input:** {{Question}}        **Output:** {{Evaluation Explanation}}
       {{Assistant 1's Response}}        {{Score of Assistant 1}}
       {{Assistant 2's Response}}        {{Score of Assistant 2}}

Figure 15: The prompt of FairEval

