# OpenReview forum: "WaAgents: A Waterfall-Inspired Framework for Effective Multi-Agent Collaboration"
_ICLR.cc/2026/Conference — ICLR 2026 Conference Withdrawn Submission_

### Official Review · Reviewer_XYWy · 2025-10-22

**Soundness:** 1
**Presentation:** 2
**Contribution:** 2
**Rating:** 0
**Confidence:** 5

**Summary:**

This paper proposes WaAgents, a multi-agent framework inspired by the Waterfall Model from software engineering. The system divides problem-solving into four sequential stages: Requirement Analysis, Decomposes tasks, assigns to worker agents, generates pseudocode
Implementation, Converts pseudocode to executable Python code, and Reflection. The authors evaluate on MGSM (math), BFCL (tool use), and MT-Bench (open tasks) using GPT-3.5-Turbo and GPT-4o.

**Summary:** A heuristic approach with mixed empirical results, no theoretical foundations, and unclear practical advantages given computational costs.

**Strengths:**

The work aims to address a real-world planning problem.  The paper has very limited strengths:

1. Clear Problem Identification, discussing real issues and proposing solutions.

2. Structured Representation: Using Gherkin format reduces ambiguity in requirement analysis.

3. Honest Limitations: Authors acknowledge poor performance with weaker models and recognize need for broader evaluation

**Weaknesses:**

This paper requires substantial enhancements to submit to a top-tier forum.

**1. Missing a number of related work** published in the multi-agent orchestration area.  Notably, the book of Multi-Agent Collaborative Intelligence (2024).  The SagaLLM paper (VLDB 2025), which deals with workflow design from problem decomposition,  workflow specification, agent specification, to coding.  The authors should also go back to those transaction processing work published in 1980s to understand some real issues and solid solutions.

**2. Arbitrary Four-Stage Decomposition** is heuristic and problematic.  No principled justification for exactly four stages. Waterfall Model is outdated and criticized in software engineering for being too rigid.  Why not three stages for real-world planning problems? Why not six? No theoretical basis provided to justify.

**3. Problematic Evaluation** which uses LLM-based evaluation (FairEval) to judge LLM outputs. Can this create circulate problem?

**4. Lacking Failure Analysis.**  What happens when verification loops fail repeatedly? Missing analysis of when the approach breaks down.

**5. Stage Success Measurement Unclear.**
* How do you reliably measure if each stage succeeded?
* Verification experts may themselves be unreliable.
* No metrics for intermediate stage quality...

**Questions:**

**1.** Why do you believe the Waterfall Model fits general problem solving perfectly?
 Software engineering abandoned this model decades ago due to its rigidity and poor fit for iterative development.

* Why assume sequential stages work for diverse problem types?
* Why exactly four stages rather than three or six? What principles guided this choice?

**2.** How do you reliably measure "success" at each stage, and how do you assess the reliability of these measures? Verification experts are themselves LLMs that may be unreliable.

* What is the error rate at each verification stage?
* How do you prevent cascading errors when early-stage verification fails?
* What happens when verification loops indefinitely?

**3.** Can you cast a wider net to survey related work, especially the MACI book (2024) and its published chapters at ICML and VLDB?
MACI is a 17-chapter book dealing comprehensively with LLM multi-agent problems.

* How does WaAgents compare to MACI's formal frameworks?
* How does it compare to SagaLLM (VLDB 2025) and transaction processing systems?
* What about decades of distributed systems and workflow research?

**4.** How does computational cost compare to baselines, and is this approach practical?

* How many total LLM calls per problem?
* What are the latency and dollar costs compared to simpler methods?
* When is the overhead justified by improved results?

**5.** Why does your approach perform worse than simpler methods on some tasks, and when should it be used?
WaAgents gets 64.8% on MGSM/GPT-3.5 vs. Camel's 84.4%

* What characteristics make a problem suitable for rigid stage decomposition?
* When does structure help vs. hurt?
* What are the failure modes and convergence guarantees?

---

### Official Review · Reviewer_AKBk · 2025-10-30

**Soundness:** 3
**Presentation:** 3
**Contribution:** 3
**Rating:** 4
**Confidence:** 3

**Summary:**

This paper proposes a novel multi-agent collaboration framework that incorporates the waterfall model from traditional software engineering. The framework introduces a four-stage pipeline for task decomposition—Requirement Analysis, Design, Implementation, and Reflection—assigning each agent a clearly defined objective. The proposed method demonstrates strong performance across multiple benchmark evaluations.

**Strengths:**

This method resolves two major issues commonly observed in multi-agent collaboration:
1. Information redundancy caused by the absence of clearly defined tasks;
2. Decision conflicts arising from uncontrolled free-form discussions.
By explicitly defining task objectives and processing them sequentially, the proposed approach enforces structured coordination among agents, thereby ensuring focused generation and consistent decision-making.

By dividing the code generation process into two distinct stages—Design and Implementation, each governed by different logical processes—the framework effectively reduces the workload and cognitive burden of a single agent.
In the Reflection Stage, the design of the Error Analysis Agent enables the system to identify and correct errors made in previous stages, thereby enhancing the overall robustness of the framework.

The paper is well-written and flows smoothly, making it easy for readers to follow the ideas and reasoning.
The experiments show strong performance on various benchmarks.

**Weaknesses:**

Although the introduction of the waterfall model and its sequential execution brings several advantages, it also reduces the framework’s flexibility. As a result, the system may experience performance degradation when handling tasks in complex real-world environments.
The framework lacks evaluation on more generalized or diverse tasks, limiting the assessment of its broader applicability.

**Questions:**

In the Requirement Analysis stage, the design that relies on active questioning is dependent on the quality of user interaction. This raises a concern about whether such interactions might cause interruptions or introduce noise and contradictions into the workflow.

The workflow is complex, involving transformations and handoffs across multiple intermediate artifacts—user logs, Gherkin specifications, pseudocode, executable code, execution logs, and review outcomes. Does this design excessively consume the LLM’s context window (token budget)? In parallel, does it introduce additional latency? Could you provide comparative measurements of reaction time?

Does the current worker profile design pose challenges for adapting to novel tasks? A static pool of experts may fail to cover fine-grained or emerging subdomains, potentially limiting specialization and task–expert matching.

---

### Official Review · Reviewer_iGmw · 2025-11-01

**Soundness:** 3
**Presentation:** 3
**Contribution:** 3
**Rating:** 6
**Confidence:** 4

**Summary:**

WaAgents introduces a waterfall-inspired multi-agent framework that turns open-ended problems into a four-stage pipeline: Requirement Analysis, Design, Implementation, and Reflection. Each stage produces a verifiable artifact (Gherkin specifications, a task plan with worker assignment, pseudocode, executable code and logs). With stage-locked deliverables and explicit failure attribution to design or implementation, the system reduces redundant dialogue and enables targeted repair. Experiments on MGSM, BFCL, and MT-Bench with GPT-3.5 Turbo and GPT-4o show strong gains, and ablations indicate that every stage contributes. Overall, WaAgents reframes multi-agent collaboration as an auditable sequence of artifacts rather than free-form debate, improving reliability and traceability.

**Strengths:**

- The paper reframes multi-agent collaboration as a waterfall pipeline with verifiable artifacts (Gherkin requirements, pseudocode, code), creatively importing software engineering structure into LLM coordination. The explicit attribution of failures to design or implementation is a clear conceptual contribution.

- The study covers three benchmarks and two model strengths with both automatic and human judgments, showing consistently strong results with GPT-4o. Stage-wise ablations demonstrate that each component is necessary.

- A concise end-to-end diagram and stage-specific prompts make the workflow transparent and reproducible. The artifact-centric presentation improves traceability and auditing.

- The method achieves state-of-the-art or near-state-of-the-art performance on reasoning and tool-use tasks while reducing coordination overhead. It provides a practical recipe for deploying reliable agent systems in real applications.

**Weaknesses:**

- The paper asserts that stage-driven collaboration “substantially reduces information redundancy” and “fundamentally eliminates decision conflicts,” yet it reports no process metrics such as turn counts, divergence episodes, or redundancy ratios to substantiate these claims.

- On MGSM with GPT-3.5-Turbo, WaAgents underperforms Camel and AgentVerse, suggesting the method depends on stronger base models to realize its benefits and may be less suitable for short reasoning chains.

- There is no accounting of token usage, latency, number of tool calls, or cost per solved task, leaving the practical efficiency of a four-stage pipeline unclear relative to baselines.

- Reproducibility gaps around Worker Profile. The approach relies on curated worker capabilities and tools, but the paper does not enumerate the full worker set, their prompts, or permissions, making fairness and replication difficult to assess.

- Reflection is described as initiated when an “anomaly” occurs, but anomaly types and thresholds are not operationalized, which limits predictability and analysis of failure modes.

**Questions:**

- How is the Reflection trigger defined in practice; what anomaly types and thresholds are used, and what is the accuracy of attributing errors to Design versus Implementation?

- Could you report token, latency, and monetary cost per task and compare methods under equal budget to quantify efficiency versus quality trade-offs?

- Do results generalize to open-source LLMs of different sizes; if not yet tested, which components break first when model capability drops?

---

### Official Review · Reviewer_HLzQ · 2025-11-01

**Soundness:** 2
**Presentation:** 1
**Contribution:** 2
**Rating:** 2
**Confidence:** 4

**Summary:**

The paper proposes WaAgents, a waterfall-inspired multi-agent framework dividing problem solving into four ordered stages: Requirement Analysis, Design, Implementation, and Reflection. Each phase uses specialized agents (e.g., clarifier, pseudocode generator, error analyst) to reduce dialogue redundancy and enable targeted debugging. Experiments on MGSM, BFCL, and MT-Bench show gains over selected baselines, particularly with stronger LLMs (GPT-4o).

**Strengths:**

1. Introduces explicit staged orchestration inspired by SE Waterfall model. Separation of requirement clarification, planning, execution, and error correction improves controllability.

2. Ablations indicate each stage contributes to performance improvement. Provides concrete prompts and engineering details, aiding reproducibility.

3. Addresses known issues in multi-agent systems: redundant discussion, conflict loops, and debugging difficulty.

**Weaknesses:**

1. **Limited scientific novelty; engineering-driven design**

- The core idea adapts a classical SE Waterfall pipeline, applying it to LLM agents feels like a process-engineering pattern rather than a conceptual advancement in multi-agent intelligence.

- The paper does not show that this staged paradigm unlocks fundamentally new reasoning capabilities beyond improved organization. Limits this work’s research contribution and makes the results vulnerable to stronger baselines with different orchestration.

2. **Model selection limits generalization claims**

- Evaluation uses GPT-3.5-Turbo and GPT-4o, which are outdated.

- Lacks experiments on open-source LLMs and diverse model families/sizes(Claude 3.5 Sonnet, DeepSeek), making transferability and scalability claims uncertain.

3. **Benchmark coverage insufficient**

- Only MGSM, BFCL, and MT-Bench are used. Missing real-world, long-horizon, or software-engineering benchmarks (e.g., SWE-Bench, MBPP, LiveCodeBench), which are natural fits for this SE-inspired framework.

4. **Baselines not state-of-the-art**

- Compared only to Camel, AgentVerse, AutoAgents, which are early frameworks. Absent stronger baseline(e.g. AutoGen, MetaGPT,), or planning-centric(XAgent)/graph-based orchestration baselines.

5. **Weak evaluation methodology**

- Only 3 annotators (master students), no inter-annotator agreement.
- No quality control: How are disagreements resolved? What is variance across annotators?

6. **No efficiency/compute analysis**

What is the total token/cost/latency overhead of WaAgents vs baselines? Please provide complete efficiency metrics.

7. **Presentation and clarity issues**

- Figure 1 extremely dense and difficult to parse
- Inconsistent terminology: "Error Analysis Agent" vs "Error Analyst"

**Questions:**

1. Quantitative evidence: The paper claims to reduce "information redundancy" and eliminate "decision conflicts." Can you provide quantitative metrics? (e.g., average dialogue turns, token efficiency, conflict rate measurement)

2. Inter-annotator agreement: What is the agreement among three human evaluators?  How are disagreements resolved and quality control enforced?

3. Failure analysis: Can you provide examples where WaAgents fails? What are common failure patterns? When does the Waterfall structure break down?

4. Generalization: Results only use GPT-3.5/4o. Have you tested on open-source models (Llama, DeepSeek, Qwen)? How does the framework degrade with weaker models?

5. How does WaAgents compare to strong baseline AutoGen, MetaGPT, and more recent planning-driven/memory-driven/debated-based multi-agent frameworks?

6. What is the token cost / latency overhead vs. discussion-based agents?

7. Please evaluate on SWE-Bench, MBPP, LiveCodeBench, or repository-level tasks to validate the hypothesis that waterfall structure is especially advantageous in SE-like workflows.

8. Could WaAgents be extended to adaptive or iterative process flows rather than fixed waterfall stages?

---

### Note · Authors · 2026-01-11

I have read and agree with the venue's withdrawal policy on behalf of myself and my co-authors.